# Adverse pregnancy outcomes among women in Norway with gestational diabetes using three diagnostic criteria

Anam Shakil Rai[1]*, Line Sletner[2,3], Anne Karen Jenum[4], Nina Cecilie Øverby[5], Signe Nilssen Stafne[6,7], Elisabeth Qvigstad[3,8], Are Hugo Pripp[9], Linda Reme Sagedal[1,10]

1 Department of Research, Sorlandet Hospital, Kristiansand, Norway, 2 Department of Pediatric and Adolescents Medicine, Akershus University Hospital, Nordbyhagen, Akershus, Norway, 3 Institute of Clinical Medicine, University of Oslo, Oslo, Norway, 4 Department of General Medicine, General Practice Research Unit (AFE), Institute of Health and Society, University of Oslo, Oslo, Norway, 5 Department of Nutrition and Public Health, Faculty of Health and Sport Sciences, University of Agder, Kristiansand, Norway, 6 Department of Public Health and Nursing, Norwegian University of Science and Technology (NTNU), Trondheim, Norway, 7 Department of Clinical Services, St.Olavs Hospital Trondheim University Hospital, Trondheim, Norway, 8 Department of Endocrinology, Morbid Obesity and Preventive Medicine, Oslo University Hospital, Oslo, Norway, 9 Oslo Centre of Biostatistics and Epidemiology, Research Support Services, Oslo University Hospital, Oslo, Norway, 10 Department of Obstetrics and Gynaecology, Sorlandet Hospital, Kristiansand, Norway

* anamsr@uia.no

**Data Availability Statement:** The datasets generated and/or analyzed during the current study are not publicly available due to the dataset containing potentially sensitive data. The editors

## Abstract

### Introduction

The aim of this study was to examine the risk of adverse perinatal outcomes in women diagnosed with GDM by the World Health Organization (WHO) 1999 criteria, and in those retrospectively identified by the Norwegian-2017 and WHO-2013 criteria but not by WHO-1999 criteria. We also examine the effect of maternal overweight/obesity and ethnicity.

### Material and methods

We used pooled data from four Norwegian cohorts (2002–2013), encompassing 2970 mother-child pairs. Results from universally offered 75-g oral glucose tolerance tests measuring fasting plasma glucose (FPG) and 2-hour glucose (2HG) were used to assign women into three diagnostic groups: Diagnosed and treated by WHO-1999 (FPG$\geq$7.0 or (2HG $\geq$7.8 mmol/L), identified by WHO-2013 (FPG $\geq$5.1 or 2HG $\geq$8.5 mmol/L), and identified by Norwegian-2017 criteria (FPG $\geq$5.3 or 2HG $\geq$9.0 mmol/L). Perinatal outcomes included large-for-gestational-age (LGA) infants, cesarean section, operative vaginal delivery, preterm birth and preeclampsia.

### Results

Compared to the non-GDM group, women diagnosed with GDM by either of the three criteria had an increased risk of large-for-gestational-age infants (adjusted odds ratios (OR) 1.7–2.2). Those identified by the WHO-2013 and Norwegian-2017 criteria but not diagnosed and treated by WHO-1999 criteria had an additional increased risk of cesarean section (OR

can access data (in de-identified form) used in the manuscript, code book, and analytical code upon request. The project manager will contribute to the access being provided under appropriate conditions. However, research data for this publication include identifying health information subject to confidentiality. It is therefore not possible to share raw data publicly. Name of ethics committee: Regional Committees for Medical and Health Research Ethics Non-author contact: The Norwegian Centre for Research Data and Anja Maria Lyche Brænd (a.m.l.brand@medisin.uio.no).

**Funding:** This work was funded by South-Eastern Norway Regional Health Authority. The funders had no role in study design, data collection and analysis, decision to publish, or preparation of the manuscript.

**Competing interests:** The authors have declared that no competing interests exist.

1.36, 95% CI 1.02,1.83 and 1.44, 95% CI 1.03,2.02, respectively) and operative vaginal delivery (OR 1.35, 95% CI 1.1,1.7 and 1.5, 95% CI 1.1,2.0, respectively). The proportions of LGA neonates and cesarean section were higher for women with GDM in both normal-weight and overweight/obese women. Asians had a lower risk of delivering large-for-gestational-age infants than Europeans applying national birthweight references, but maternal glucose values were similarly positively associated with birthweight in all ethnic groups.

## Conclusions

Women who met the WHO-2013 and Norwegian-2017 criteria, but were not diagnosed by the WHO-1999 criteria and therefore not treated, had an increased risk of LGA, cesarean section and operative vaginal delivery compared to women without GDM.

## Introduction

Gestational diabetes mellitus (GDM) is associated with increased risk of macrosomia, cesarean section, preeclampsia and preterm delivery [1], and with long term increased risk of obesity and type 2 diabetes in both mother and child [2].

Diagnostic thresholds of GDM applied in Norway were previously derived from criteria for glucose intolerance used for non-pregnant individuals [3]. In 2013, the World Health Organization (WHO) recommended glycaemic thresholds for the diagnosis of GDM based on findings from the multinational Hyperglycemia and Adverse Pregnancy Outcome (HAPO) study, demonstrating a linear dose-response between maternal glycaemia and adverse neonatal outcomes [4]. These criteria identified women with an adjusted odds ratio (OR) of 1.75, relative to the mean, for pre-specified outcomes, such as large-for-gestational-age (LGA) neonates, primary cesarean section, and neonatal hypoglycaemia [5]. Glucose values set to identify women with a higher risk were also considered but rejected [4]. Nonetheless, several countries, including Canada, Finland, and Norway, adopted thresholds corresponding to a 2-fold risk for these outcomes.

Shifting from the former WHO-1999 criteria to the WHO-2013 criteria has been shown to increase the prevalence of GDM considerably due to a higher case identification of women with moderately increased fasting glucose levels only [6, 7]. However, it is unclear whether women classified as GDM by the new criteria but as non-GDM previously, have a clear risk of adverse pregnancy outcomes with a magnitude that warrants treatment, and whether ethnic background and overweight/obesity influence these relationships.

Accordingly, we aimed to explore the risk for LGA, cesarean delivery, operative vaginal delivery, preterm birth, and preeclampsia in women i) identified and treated for GDM by the WHO-1999 criteria or ii) identified by the Norwegian-2017 and the WHO-2013 criteria, but not by the WHO-1999 criteria (and therefore not treated), also taking maternal overweight/obesity and ethnicity into account.

## Material and methods

### Study design and population

We used data from The Norwegian Hyperglycaemia in Pregnancy consortium, a merged data set with two cohort studies [8, 9] and two randomized controlled trials (RCT) [10, 11] conducted in Norway between 2002 and 2013. The interventions in the two trials consisted of

either an exercise program or a combination of a physical activity component and dietary counselling, but these interventions demonstrated no effect on GDM incidence or LGA and cesarean section. The four studies were merged to perform a pooled analysis. Detailed study methods for the pooled data set have been previously described [12] and participant characteristics for all studies are summarized in S1 Table in S1 File. In short, included studies comprised women with singleton live-born neonates recruited early in pregnancy (between week 15–20 with data on maternal age and pre-pregnancy BMI, glucose measurements obtained from at least one universally offered 75g 2-hour oral glucose tolerance test (OGTT) performed ≥ 20 weeks' gestation, and at least one offspring measurement (birthweight). Only studies that had core data and were not based on specific selection criteria (e.g. obese women only) were considered for inclusion.

## Obstetric and neonatal outcomes

Data on obstetric outcomes were obtained from hospital records, including mode of delivery (normal vaginal delivery, total cesarean section (planned or emergency), operative vaginal delivery (vacuum extraction or forceps)), gestational age at birth, preeclampsia or severe hypertensive disorder, and preterm delivery (<37 weeks of pregnancy). Routine anthropometric measurements (birthweight and length) of neonates were performed by study staff immediately following birth. Birthweight z-score and large-for-gestational-age (LGA) (birthweight > 90th percentile) were calculated using Norwegian sex and gestational age-specific national references [13]. Birth weight z-scores express the weight as the number of standard deviations (SD) above or below the reference mean value for a specific gestational age and sex (not customized for ethnicity, maternal height etc.)

## Main exposure and covariates

Our main exposure variable was GDM. During the data collection, the diagnosis of GDM was based exclusively on the WHO-1999 criteria. If diagnosed, women received standard GDM care according to national guidelines, which remained unchanged during the data collection period, with self- monitoring of blood glucose and dietary counselling. Oral antidiabetic therapy or insulin was commenced if blood glucose levels repeatedly exceeded treatment targets. Only 12 women received such pharmacological treatment.

We additionally applied the WHO-2013 diagnostic cut-offs (only fasting and 2-hour values as 1-hour glucose was not measured in these studies) and the Norwegian-2017 cut-offs to the same diagnostic OGTT. Based on their OGTT results, women were retrospectively assigned to the following (partly overlapping) diagnostic groups:

1. GDM diagnosed and treated according to WHO-1999 criteria (fasting glucose ≥7.0 mmol/l and/or 2-hour glucose (2HG) ≥7.8 mmol/l).

2. GDM retrospectively identified according to WHO-2013 criteria (fasting glucose ≥5.1 mmol/l and/or 2HG ≥8.5 mmol/l).

3. GDM retrospectively identified according to Norwegian-2017 criteria (fasting glucose ≥5.3 mmol/l and/or 2HG ≥9.0 mmol/l).

All participants provided questionnaire data, self-reported [9–11] or through interviews [8]. Height was measured directly at sites while weight prior to pregnancy was self-reported. Pre-pregnancy body mass index (BMI) was calculated and categorized according to the WHO International Classification of normal weight (≤24.9 kg/m$^2$), overweight (25–29.9 kg/m$^2$) and obesity (≥30 kg/m$^2$).

Ethnic origin was defined by the pregnant woman's mother's country of birth and further merged into three groups in the current study: European (predominantly Scandinavian as well as East and West-European origin), Middle Eastern/African, and Asian (primarily South and East Asian ethnicity) [8].

The four birth cohorts provided data from 3315 pregnant women and 3293 live births (S1 Fig). After excluding women with multiple pregnancies, those lacking glucose values, infants with missing birthweight and fetal deaths the study sample consisted of 2970 mother-child pairs.

## Statistical analyses

Assumptions for statistical analysis were tested and distributions of all potential covariates were checked for normality using Tests of Normality and inspection of probability plots, which confirmed that these variables followed a normal distribution. Data are reported as frequencies and percentages for categorical variables and mean and standard deviation for continuous variables, using $X^2$ test or Student's $t$ Test as appropriate.

To assign values for the missing data for pre-pregnancy weight (5%), height (0.4%), educational attainment (0.3%) and parity (0.3%) we used Stochastic regression imputation with predictive mean matching as the imputation model. Statistical analyses were carried out using statistical package IBM SPSS (version 23.0. Armonk, NY: IBM Corp).

Logistic regression models were used to estimate the OR and 95% confidence intervals (CI) for associations between maternal GDM status and clinical outcomes before and after adjustment for maternal age, pre-pregnancy BMI, ethnicity, parity, smoking and gestational age at birth. We also adjusted for study cohort to handle potential unmeasured confounders. We did not further adjust for maternal education, as it was not associated with our perinatal outcomes, and adding this variable to the models had no impact on the effect estimates of interest. During data collection, information about GDM and subsequent treatment was only offered to women diagnosed according to the WHO-1999 criteria. Hence, in the final models assessing the effect of GDM by Norwegian-2017 and WHO-2013 criteria, we additionally adjusted for whether they were diagnosed and offered treatment for GDM by the WHO-1999 criteria. Doing so allowed us to identify the group of women with an elevated fasting blood glucose only (fasting glucose 5.1–6.9 mmol/l and 2HG <7.8 mmol/l for WHO-2013, and fasting glucose 5.3–6.9 mmol/l and 2HG <7.8 mmol/l for Norwegian-2017 criteria) who were untreated. As a sensitivity analysis, and to verify the results achieved by the analyses where adjustment for treatment and a known diagnosis were made to the model, we repeated the same analysis after excluding participants who were diagnosed and treated based on the WHO-1999 criteria. Results are presented as unadjusted and multivariable-adjusted models.

As the definition used for LGA was derived from a predominantly ethnic Norwegian population, we explored in separate general linear models the effect of maternal glucose values on offspring birthweight z-score, stratified by ethnic groups in adjusted models. The conditions for a linear regression were checked, confirming a linear relationship between maternal glucose values and offspring birthweight. We did not mutually adjust for FPG and 2HG due to collinearity. All $p$ values are two-tailed, and $p$ values <0.05 were considered statistically significant.

## Ethics

All studies were based on written informed consent. The Norwegian Regional Ethics committees (REC) approved that each constituent study could contribute to the consortium, and the current study was approved by the REC South East (2017/2533).

## Results

Characteristics and pregnancy outcomes of the total cohort stratified by GDM status are presented in Table 1. In total 10.7% of women were diagnosed with GDM based on the WHO-1999 criteria, 16.9% with WHO-2013, and 10.3% with Norwegian-2017 criteria. As expected, women with GDM were older and had a higher pre-pregnancy BMI than those without GDM. All three groups of women diagnosed with GDM (according to WHO-1999, WHO-2013 and Norwegian-2017 criteria) had a higher rate of LGA neonates compared to their non-GDM counterparts, while higher rates of macrosomia (birthweight>4000g) were only found in those diagnosed by the Norwegian-2017 and WHO-2013 criteria. Similarly, women in all GDM groups had an increased risk of cesarean section, but only those who met the Norwegian-2017 and WHO-2013 criteria were more likely to have an operative vaginal delivery compared to their non-GDM counterparts. Only women diagnosed with GDM by the WHO-1999 criteria had a higher risk of preterm birth and preeclampsia.

Fig 1 shows relations between GDM and (a) LGA and (b) cesarean section in women with normal-weight or overweight/obesity. GDM by any criteria significantly increased the proportion of LGA infants in both normal-weight and overweight/obese subgroups, although the highest proportions were observed for overweight/obese women (S2 Table in S1 File). Similar results were found for cesarean section except when applying the WHO-1999 criteria.

The unadjusted and adjusted associations between GDM and LGA, cesarean section and operative vaginal delivery for each of the three GDM criteria are reported in Table 2. For those diagnosed with GDM according to the WHO-1999 criteria and treated accordingly, an increased risk was only found for delivering an LGA infant (adjusted OR 2.22, 95% CI 1.5,3.2). Women retrospectively classified as having GDM by the WHO-2013 criteria had a crude OR of 2.08 for LGA. After adjusting for confounders and for treatment by the WHO-1999 criteria (thereby expressing the risk related to having fasting glucose 5.1–6.9 mmol/l while 2HG <7.8 mmol/l) the OR for LGA for this group was 1.70 (95% CI 1.2,2.5,) compared to non-GDM women (both fasting glucose <5.1 mmol/l and 2HG <7.8 mmol/l). These women also had an increased risk of total cesarean section (OR 1.36, 95% CI 1.02,1.83). and for operative vaginal delivery (OR 1.35, 95% CI 1.1,1.7). Women identified with the Norwegian-2017 but not the WHO-1999 criteria (i.e. fasting glucose 5.3–6.9 mmol/l while 2HG ≤7.8) compared to women not identified by any of the two criteria, had an increased adjusted risk of LGA (OR 2.05, 95% CI 1.3,3.1), cesarean section (OR 1.44, 95% CI 1.03,2.02), operative vaginal delivery (OR 1.50, 95% CI 1.1,2.0) and emergency cesarean section (OR 1.57, 95% CI 1.1,2.3, P = 0.024).

Sensitivity analyses (Table 3), applied on women with GDM by the WHO-2013 and Norwegian-2017 criteria after excluding those with GDM by the WHO-1999 criteria, showed almost identical effect estimates although statistical significance was reached for LGA only, presumably due to decreased population size. There was no significantly increased risk of preterm delivery or preeclampsia regardless of GDM criteria applied (S3 Table in S1 File).

Asians had substantially lower risk of delivering an LGA infant than Europeans (Table 2), however, as the definition of LGA in Norway was not ethnicity-specific, we further explored the linear association between maternal glucose and birthweight z-score separately for Europeans, South Asians, and Middle Eastern/Africans (Table 4). Fasting glucose was significantly associated with higher birthweight in all ethnic groups, implying that one mmol/l increase in fasting glucose was associated with an increase in birthweight z-score by 0.30 SD units after adjustments for relevant covariates, equivalent to approximately 130g in a full-term neonate. Similarly, 2HG was positively associated with birthweight z-score in all groups in both univariable simple and multivariable adjusted analyses.

**Table 1. Characteristics and pregnancy outcomes in total study sample and according to their glucose tolerance status, using three diagnostic criteria for gestational diabetes.** For each column data are presented as mean ± SD or n (%).

| | Total cohort | WHO-1999 criteria | | | WHO-2013 criteria | | | Norwegian-2017 criteria | | |
|---|---|---|---|---|---|---|---|---|---|---|
| *Participant characteristic* | *2970* | non-GDM, n = 2652 (89.3) | GDM, n = 318 (10.7) | P[a] Value | non-GDM, n = 2469 (83.1) | GDM, n = 501 (16.9) | P[a] Value | non-GDM, n = 2663 (89.7) | GDM, n = 307 (10.3) | P[a] Value |
| Maternal age (years) | 30.0 (4.4) | 29.9 ± 4.4 | 31.6 ± 4.6 | < .001 | 30.0 ± 4.3 | 30.8 ± 5.0 | < .001 | 30.1 ± 4.3 | 30.8 ± 5.1 | .007 |
| Pre-pregnancy BMI (kg/m²) | 23.7 ± 3.9 | 23.6 ± 3.9 | 24.8 ± 4.5 | < .001 | 23.3 ± 3.5 | 25.7 ± 5.1 | < .001 | 23.4 ± 3.6 | 26.0 ± 5.5 | < .001 |
| Pre-pregnancy BMI groups (kg/m²), n (%) | | | | < .001 | | | < .001 | | | < .001 |
| Normalweight ≤24.9 | 2127 (71.7) | 1948 (73.5) | 179 (56.3) | | 1864 (75.5) | 263 (52.5) | | 1974 (74.1) | 153 (49.8) | |
| Overweight 25–29.9 | 610 (20.5) | 516 (19.5) | 94 (29.6) | | 462 (18.7) | 148 (29.5) | | 524 (19.7) | 86 (28.0) | |
| Obesity ≥30 | 233 (7.8) | 188 (7.1) | 45 (14.2) | | 143 (5.8) | 90 (18.0) | | 165 (6.2) | 68 (22.1) | |
| Ethnicity, n (%) | | | | .055 | | | < .001 | | | < .001 |
| European | 253 (86.6) | 2311 (87.1) | 262 (82.4) | | 2221 (90.0) | 352 (70.3) | | 2373 (89.1) | 200 (65.1) | |
| Middle-Eastern/African | 174 (5.9) | 151 (5.7) | 23 (7.2) | | 113 (4.6) | 61 (12.2) | | 133 (5.0) | 41 (13.4) | |
| Asian | 223 (7.5) | 190 (7.2) | 33 (10.4) | | 135 (5.5) | 88 (17.6) | | 157 (5.9) | 66 (21.5) | |
| Primipara, n (%) | 1814 (61.1) | 1621 (61.1) | 193 (60.7) | .881 | 1150 (62.8) | 264 (52.7) | < .001 | 1656 (62.2) | 158 (51.5) | < .001 |
| Education, n (%) | | | | .009 | | | < .001 | | | < .001 |
| Primary or less | 146 (4.9) | 120 (4.5) | 26 (8.2) | | 89 (3.6) | 57 (11.4) | | 105 (3.9) | 41 (13.4) | |
| High school education | 640 (21.5) | 566 (21.3) | 74 (23.3) | | 493 (20.0) | 147 (29.3) | | 544 (20.4) | 96 (31.3) | |
| Higher education | 2184 (73.5) | 1966 (74.1) | 218 (68.6) | | 1887 (76.4) | 297 (59.3) | | 2014 (75.6) | 170 (55.4) | |
| Current smoker, n (%) | 80 (2.8) | 72 (2.8) | 8 (2.7) | .885 | 62 (2.6) | 18 (3.9) | .116 | 68 (2.7) | 12 (4.3) | .108 |
| Fasting glucose at OGTT (mmol/L) | 4.6 ± 0.5 | 4.5 ± 0.4 | 5.0 ± 0.6 | < .001 | 4.4 ± 0.3 | 5.3 ± 0.5 | < .001 | 4.5 ± 0.4 | 5.5 ± 0.6 | < .001 |
| 2-hour glucose at OGTT (mmol/L) | 6.1 ± 1.3 | 5.7 ± 1.0 | 8.6 ± 0.8 | < .001 | 5.8 ± 1.1 | 7.4 ± 1.6 | < .001 | 5.9 ± 1.2 | 7.6 ± 1.7 | < .001 |
| Gestational age at OGTT (weeks) | 30.8 ± 2.5 | 30.8 ± 2.5 | 30.5 ± 2.3 | .005 | 31.0 ± 2.5 | 29.8 ± 2.2 | < .001 | 31.0 ± 1.7 | 31.0 ± 2.5 | < .001 |
| Gestational age at delivery (weeks) | 39.8 (1.6) | | | | | | | | | |
| *Outcome* | | | | | | | | | | |
| Birthweight, gram | 3520 (522) | 3517.9 (518.0) | 3537.7 (555.2) | .523 | 3505 (515.7) | 3594 (547) | < .001 | 3512 (517) | 3588 (557) | .016 |
| LGA, n (%) | 230 (7.7) | 184 (6.9) | 46 (14.5) | < .001 | 165 (6.7) | 65 (13.0) | < .001 | 183 (6.9) | 47 (15.3) | < .001 |
| Birthweight z-score | - 0.05 (0.9) | -0.748 (0.93) | 0.085 (1.00) | .004 | -0.097 (0.92) | 0.138 (1.01) | < .001 | -0.082 (0.92) | 0.158 (1.15) | < .001 |
| Macrosomia ≥4000g, n (%) | 507 (17.1) | 444 (16.7) | 63 (19.8) | .168 | 392 (15.9) | 115 (23.0) | < .001 | 433 (16.3) | 74 (24.1) | .001 |
| Preterm birth, n (%) | 108 (3.9) | 90 (3.4) | 20 (6.3) | .010 | 87 (3.5) | 23 (4.6) | .211 | 92 (3.5) | 16 (5.2) | .120 |
| Preeclampsia, n (%) | 98 (3.6) | 81 (3.3) | 17 (5.7) | .036 | 82 (3.7) | 16 (3.4) | .751 | 88 (3.6) | 10 (3.4) | .872 |
| Total cesarean section, n (%) | 446 (15.0) | 378 (14.3) | 68 (21.4) | .004 | 339 (13.7) | 107 (21.4) | < .001 | 375 (14.1) | 71 (23.1) | < .001 |
| emergency | 298 (10.0) | 258 (9.7) | 40 (12.6) | | 230 (9.3) | 68 (13.6) | | 250 (9.4) | 48 (15.6) | |
| planned | 148 (5.0) | 120 (4.5) | 28 (8.8) | | 109 (4.4) | 39 (7.8) | | 125 (4.7) | 23 (7.5) | |
| Operative vaginal delivery, n (%) | 386 (13.0) | 737 (27.8) | 105 (33.0) | 0.051 | 672 (27.2) | 170 (33.9) | 0.002 | 730 (27.4) | 112 (36.5) | 0.001 |

*(Continued)*

**Table 1.** (Continued)

| *Participant characteristic* | Total cohort | WHO-1999 criteria | | | WHO-2013 criteria | | | Norwegian-2017 criteria | | |
|---|---|---|---|---|---|---|---|---|---|---|
| | *2970* | non-GDM, n = 2652 (89.3) | GDM, n = 318 (10.7) | P^a Value | non-GDM, n = 2469 (83.1) | GDM, n = 501 (16.9) | P^a Value | non-GDM, n = 2663 (89.7) | GDM, n = 307 (10.3) | P^a Value |
| **Received treatment/ known diagnosis** | 318 | 2652 (0) | 318 (100) | | 119 (4.8) | 199 (39.7) | | 180 (6.8) | 138 (45.0) | |

[a]Independent sample T test for continuous variables and $X^2$ statistic for categorical variables.

WHO: World Health Organization, GDM: gestational diabetes mellitus, BMI: body mass index, OGTT: oral glucose tolerance test, LGA: large-for-gestational-age

Values are imputed for pre-pregnancy weight, parity and education.

## Discussion

Among women universally tested for GDM by OGTT, we observed that the proportion of LGA infants was significantly higher in women with GDM identified by all three criteria compared to women without GDM. Those retrospectively identified by the Norwegian-2017 and WHO-2013 criteria, but who were not diagnosed and treated by the WHO-1999 criteria (i.e., women with moderately elevated fasting glucose only) also had an increased risk of cesarean section and operative vaginal delivery after adjustment for confounders. The proportion of LGA neonates and cesarean section was higher for women with GDM in both normal-weight and overweight/obese women. Although Asian women had a reduced risk of delivering a LGA

a) Large-for-gestational-age

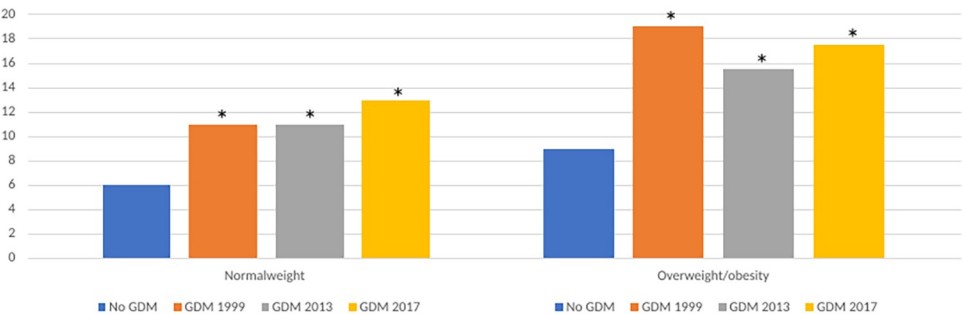

b) Cesarean section

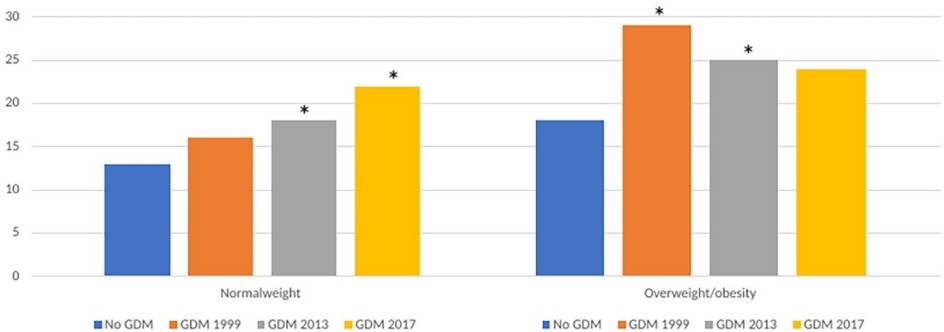

**Fig 1.** Proportion of a) large-for-gestational-age and b) cesarean section by GDM status in normal-weight and obese/overweight women. *Each GDM-category is compared with the non-GDM group using Chi-square test and significant results are marked. The non-GDM groups are represented in one single bar as the values were identical.

**Table 2. Crude and adjusted analyses of risk of large-for-gestational-age, total cesarean section and operative vaginal delivery, by GDM-criteria.**

| | Crude analysis | | | Adjusted analysis | | | | | | | | |
|---|---|---|---|---|---|---|---|---|---|---|---|---|
| | | | | WHO-1999 criteria | | | WHO-2013 criteria | | | Norwegian-2017 criteria | | |
| | Crude OR | (95% CI) | p-value | aOR* | (95% CI) | p-value | aOR∞ | (95% CI) | p-value | aOR∞ | (95% CI) | p-value |
| **Large-for-gestational-age baby** | | | | | | | | | | | | |
| GDM identified by Norwegian-2017 criteria | 2.45 | (1.7–3.5) | <0.001 | | | | | | | **2.05** | **(1.3–3.1)** | **0.001** |
| identified by WHO-2013 criteria | 2.08 | (1.5–2.8) | <0.001 | | | | **1.69** | **(1.2–2.5)** | **0.007** | | | |
| diagnosed and treated by WHO-1999 criteria | 2.26 | (1.6–3.2) | <0.001 | 2.22 | (1.5–3.2) | <0.001 | 1.73 | (1.1–2.6) | 0.009 | 1.72 | (1.1–2.6) | 0.009 |
| Prepregnancy BMI | | | | | | | | | | | | |
| Normalweight | 1 | | | 1 | | | 1 | | | 1 | | |
| Overweight | 1.59 | (1.2–2.2) | 0.004 | 1.48 | (1.1–2.1) | 0.020 | 1.42 | (1.0–2.0) | 0.038 | 1.43 | (1.0–2.0) | 0.035 |
| Obesity | 2.11 | (1.4–3.2) | <0.001 | 1.76 | (1.1–2.8) | 0.018 | 1.60 | (1.0–2.6) | 0.053 | 1.55 | (1.0–2.5) | 0.076 |
| Ethnicity | | | | | | | | | | | | |
| European ethnicty | 1 | | | 1 | | | 1 | | | 1 | | |
| Middle Eastern/African ethnicty | 1.05 | (0.6–1.8) | 0.860 | 0.69 | (0.3–1.4) | 0.326 | 0.66 | (0.3–1.4) | 0.276 | 0.65 | (0.3–1.4) | 0.251 |
| Asian | 0.15 | (0–0.5) | 0.001 | 0.11 | (0.0–0.5) | 0.003 | 0.10 | (0.0–0.4) | 0.002 | 0.09 | (0.0–0.4) | 0.001 |
| **Cesarean section (total; emergency and planned)** | | | | | | | | | | | | |
| GDM identified by Norwegian-2017 criteria | 1.83 | (1.3–2.4) | <0.001 | | | | | | | **1.44** | **(1.0–2.0)** | **0.033** |
| identified by WHO-2013 criteria | 1.70 | (1.3–2.1) | <0.001 | | | | **1.36** | **(1.0–1.8)** | **0.037** | | | |
| diagnosed and treated by WHO-1999 criteria | 1.63 | (1.2–2.1) | 0.001 | 1.19 | (0.9–1.6) | 0.262 | 1.02 | (0.7–1.4) | 0.903 | 1.04 | (0.7–1.4) | 0.807 |
| Prepregnancy BMI | | | | | | | | | | | | |
| Normalweight | 1 | | | 1 | | | 1 | | | 1 | | |
| Overweight | 1.49 | (1.2–1.9) | 0.001 | 1.47 | (1.1–1.9) | 0.002 | 1.43 | (1.1–1.8) | 0.005 | 1.44 | (1.1–1.9) | 0.004 |
| Obesity | 1.77 | (1.2–2.5) | 0.001 | 1.74 | (1.2–2.5) | 0.002 | 1.65 | (1.2–2.4) | 0.006 | 1.64 | (1.1–2.3) | 0.007 |
| Ethnicity | | | | | | | | | | | | |
| European ethnicty | 1 | | | 1 | | | 1 | | | 1 | | |
| Middle Eastern/African ethnicty | 1.00 | (0.6–1.5) | 1.000 | 0.84 | (0.5–1.4) | 0.508 | 0.82 | (0.5–1.4) | 0.443 | 0.81 | (0.5–1.4) | 0.428 |
| Asian | 1.13 | (0.7–1.6) | 0.509 | 0.95 | (0.6–1.5) | 0.841 | 0.91 | (0.6–1.4) | 0.683 | 0.90 | (0.6–1.4) | 0.643 |
| **Operative vaginal delivery** | | | | | | | | | | | | |
| GDM identified by Norwegian-2017 criteria | 1.52 | (1.2–1.9) | 0.001 | | | | | | | **1.50** | **(1.1–2.0)** | **0.006** |
| identified by WHO-2013 criteria | 1.37 | (1.1–1.7) | 0.002 | | | | **1.35** | **(1.1–1.7)** | **0.017** | | | |
| diagnosed and treated by WHO-1999 criteria | 1.28 | (0.9–1.6) | 0.051 | 1.07 | (0.8–1.4) | 0.604 | 0.93 | (0.7–1.2) | 0.614 | 0.93 | (0.7–1.2) | 0.620 |
| Prepregnancy BMI | | | | | | | | | | | | |
| Normalweight | 1 | | | 1 | | | 1 | | | 1 | | |
| Overweight | 1.27 | (1–1.5) | 0.018 | 1.29 | (1.1–1.6) | 0.013 | 1.26 | (1.0–1.5) | 0.024 | 1.27 | (1.0–1.6) | 0.021 |
| Obesity | 1.43 | (1.1–1.9) | 0.014 | 1.49 | (1.1–2.0) | 0.009 | 1.42 | (1.0–1.9) | 0.025 | 1.40 | (1.0–1.9) | 0.031 |
| Ethnicity | | | | | | | | | | | | |
| European ethnicty | 1 | | | 1 | | | 1 | | | 1 | | |
| Middle Eastern/African ethnicty | 0.95 | (0.7–1.3) | 0.790 | 1.22 | (0.8–1.9) | 0.348 | 1.19 | (0.8–1.8) | 0.420 | 1.18 | (0.8–1.8) | 0.443 |

(*Continued*)

**Table 2.** (Continued)

| | Crude analysis | | | Adjusted analysis | | | | | | | | |
| | | | | WHO-1999 criteria | | | WHO-2013 criteria | | | Norwegian-2017 criteria | | |
| | Crude OR | (95% CI) | p-value | aOR* | (95% CI) | p-value | aOR∞ | (95% CI) | p-value | aOR∞ | (95% CI) | p-value |
| Asian | 0.92 | (0.7–1.3) | 0.607 | 1.18 | (0.8–1.7) | 0.408 | 1.12 | (0.8–1.7) | 0.564 | 1.10 | (0.7–1.6) | 0.639 |

\* Adjusted for BMI group and ethnicity, as shown. Additionally adjusted for age, gestational weeks at delivery, parity and study cohort.

∞Adjusted for GDM diagnosed by WHO-1990 criteria, BMI group and ethnicity, as shown. Additionally adjusted for age, gestational weeks at delivery, parity and study cohort.

BMI categories: normalweight $\leq$24.9 kg/m$^2$, overweight 25–29.9 kg/m$^2$, obesity $\geq$30 kg/m$^2$

WHO: World Health Organization, GDM: gestational diabetes mellitus, BMI: body mass index, OR: odds ratio, aOR: adjusted odds ratio

infant compared with Europeans, a similar positive association between maternal glucose values and birthweight z-scores was found in all ethnic groups. Taken together, these findings indicate that moderately elevated fasting glucose, when unidentified and untreated, is associated with several adverse outcomes that were not observed in women where GDM was detected and treated according to WHO-1999 diagnostic criteria, which are primarily based on elevated 2HG values.

Conflicting evidence exists regarding the impact of introducing the WHO-2013 criteria on perinatal outcomes. Studies evaluating these criteria showed in general that women who would not have been identified with other criteria had higher adverse outcome rates compared to non-GDM women [7, 14, 15]. To our knowledge, only few studies, mainly from Canada, have examined adverse perinatal outcomes associated with implementing the 2.0 risk thresholds identified by the HAPO study, employed in Norwegian-2017 criteria [16–18]. One study found that when compared to women without GDM, those diagnosed with the equivalent of the Norwegian-2017 criteria had a significantly higher risk of preeclampsia, preterm birth, LGA and several other adverse outcomes, while the same was only found for LGA in the WHO-2013-only-group (fasting glucose 5.1–5.2 mmol/l and and/or 2HG 8.5–8.9 mmol/l) [17]. Although we didn't create mutually exclusive GDM categories, our finding of larger risks for all the examined outcomes when applying the Norwegian-2017 compared to WHO-2013 criteria is in line with this. However, we did not find an increased risk for preterm delivery and preeclampsia, a finding that could be related to small numbers for these outcomes in our study, indicating low power to detect associations with GDM.

As the pathophysiology of GDM is intimately linked to maternal overweight/obesity and gestational weight gain, it is often difficult to sort out the differential contributions of maternal obesity and hyperglycaemia to pregnancy outcomes. The idea that maternal BMI level is a better predictor than glucose alone for outcomes frequently associated with GDM has been widely reported in the past [19–21]. Consistent with others [22, 23], our findings indicate that, in addition to glucose levels, maternal pre-pregnancy BMI has a strong independent association with most of the examined outcomes. However, the significant increase of LGA also in normal-weight women with GDM supports a role of hyperglycaemia not attributed to maternal BMI alone.

Although we found that Asians had a reduced risk of LGA, stratified analyses of birthweight z-score suggests that the association with elevated glucose levels was similar in all ethnic groups. We have previously shown that Asians have the highest GDM prevalence irrespective of criteria used, and in particular with the WHO-2013 criteria, when compared to Europeans [12]. However, as neonates with Asian origin generally have lower birthweight compared to Europeans, our finding of a low LGA risk is not surprising as LGA was, as in most countries,

**Table 3. Adjusted analysis of risk of large-for-gestational-age, total cesarean section and operative vaginal delivery after excluding women with a GDM diagnosis based on WHO-1999 criteria.**

| n = 2652 | Norwegian-2017 criteria only[a] (n = 119) | | | WHO-2013 criteria® (n = 180) | | |
|---|---|---|---|---|---|---|
| | aOR | (95% CI) | p-value | aOR | (95% CI) | p-value |
| **Large-for-gestational-age*** | | | | | | |
| GDM | 2.02 | (1.7–3.5) | 0.012 | | | |
| Identified by Norwegian-2017 criteria | | | | | | |
| Identified by WHO-2013 | | | | 1.75 | 1.1–2.7 | 0.014 |
| Prepregnancy BMI | | | | | | |
| Normalweight | | | | | | |
| Overweight | 1.314 | 0.9–1.9 | 0.154 | 1.303 | 0.8–1.8 | 0.168 |
| Obesity | 1.697 | 0.9–2.9 | 0.057 | 1.701 | 0.9–2.9 | 0.055 |
| Ethnicity | | | | | | |
| European ethnicty | | | | | | |
| Middle Eastern/African ethnicty | 0.711 | 0.3–1.6 | 0.414 | 0.717 | 0.3–1.6 | 0.425 |
| Asian | 0.066 | 0.0–0.5 | 0.008 | 0.067 | 0.0–0.5 | 0.008 |
| **Total cesarean deliveries (emergency and elective)[b]** | | | | | | |
| GDM | 1.476 | 0.9–2.2 | 0.065 | | | |
| Identified by Norwegian-2017 criteria | | | | | | |
| Identified by WHO-2013 criteria | | | | 1.306 | 0.9–1.8 | 0.121 |
| Prepregnancy BMI | | | | | | |
| Normalweight | | | | | | |
| Overweight | 1.357 | 0.0–1.7 | 0.028 | 1.355 | 1.0–1.7 | 0.029 |
| Obesity | 1.490 | 0.9–2.2 | 0.055 | 1.5 | 0.9–2.2 | 0.051 |
| Ethnicity | | | | | | |
| European ethnicty | | | | | | |
| Middle Eastern/African ethnicty | 0.908 | 0.5–1.6 | 0.732 | 0.912 | 0.5–1.5 | 0.743 |
| Asian | 0.959 | 0.6–1.6 | 0.871 | 0.973 | 0.6–1.6 | 0.915 |
| **Operative vaginal delivery[b]** | | | | | | |
| GDM | 1.411 | 0.9–2.0 | 0.058 | | | |
| Identified by Norwegian-2017 criteria | | | | | | |
| Identified by WHO-2013 criteria | | | | 1.251 | 0.9–1.6 | 0.119 |
| Prepregnancy BMI | | | | | | |
| Normalweight | | | | | | |
| Overweight | 1.231 | 0.9–1.5 | 0.064 | 1.23 | 0.9–1.5 | 0.066 |
| Obesity | 1.351 | 0.9–1.8 | 0.112 | 1.326 | 0.9–1.8 | 0.101 |
| Ethnicity | | | | | | |
| European ethnicty | | | | | | |
| Middle Eastern/African ethnicty | 1.381 | 0.8–2.1 | 0.157 | 1.388 | 0.8–2.1 | 0.153 |
| Asian | 1.136 | 0.7–1.7 | 0.562 | 1.152 | 0.7–1.7 | 0.519 |

[a]The analysis excludes treated women n = 318, and includes women with fasting glucose 5.3–6.9 and 2-h glucose ≤7.8

®The analysis excludes treated women n = 318, and includes women with fasting glucose 5.1–6.9 mmol/l and 2HG <7.8 mmol/l

*Adjusted for age, maternal smoking, parity and study cohort in addition.

[b] Adjusted for age, gestational weeks at delivery, parity and study cohort in addition

aOR:adjusted odds ratio, GDM: gestational diabetes, WHO: Word Health Organization, BMI: body mass index, CI: confidence interval

assessed using a national reference population, rather than ethnically customised centile charts. Previous studies, including one of the pooled cohort studies, have shown that offspring of Asian women have a birthweight distribution that is skewed compared with the distribution

**Table 4. Linear regressions of maternal glucose on offspring's birthweight z-score, stratified for ethnic group.**

**European, n = 2573 (86.6%)**

| | | | Fasting glucose | | 2-hour glucose | |
| --- | --- | --- | --- | --- | --- | --- |
| | Unadjusted | | Adjusted* | | Adjusted* | |
| | β | (95% CI) | β | (95% CI) | β | (95% CI) |
| Fasting glucose[a] | **0.364** | (0.3–0.4) | **0.336** | (0.2–0.4) | | |
| 2-hour glucose[a] | **0.084** | (0.1–0.1) | | | **0.081** | (0.1–0.1) |
| Prepregnancy body mass index | | | | | | |
| Normalweight | 1 | | 1 | | 1 | |
| Overweight | **0.251** | (0.2–0.3) | **0.184** | (0.1–0.3) | **0.218** | (0.1–0.3) |
| Obesity | **0.326** | (0.2–0.5) | **0.215** | (0.1–0.4) | **0.277** | (0.1–0.4) |
| Age | **0.016** | (0.0–0.0) | **-0.010** | (-0.0-(-0.0)) | -0.009 | (-0.02–0.0) |

**Middle Eastern/African ethnicity, n = 174 (5.9%)**

| | | | Fasting glucose | | 2-hour glucose | |
| --- | --- | --- | --- | --- | --- | --- |
| | Unadjusted | | Adjusted* | | Adjusted* | |
| | β | (95% CI) | β | (95% CI) | β | (95% CI) |
| Fasting glucose[a] | **0.301** | (0.1–0.5) | **0.318** | (0.0–0.6) | | |
| 2-hour glucose[a] | **0.129** | (0.0–0.2) | | | **0.201** | (0.0–0.4) |
| Prepregnancy body mass index | | | | | | |
| Normalweight | 1 | | 1 | | 1 | |
| Overweight | 0.063 | (0.3–0.4) | -0.057 | (-0.5–0.3) | 0.018 | (-0.4–0.4) |
| Obesity | **0.609** | (0.2–0.9) | **0.496** | (0.1–0.9) | **0.592** | (0.1–1.1) |
| Age | 0.018 | (-0.01–0.0) | 0.000 | (-0.04–0.0) | -0.005 | (-0.04–0.3) |

**Asian, n = 223 (7%)**

| | | | Fasting glucose | | 2-hour glucose | |
| --- | --- | --- | --- | --- | --- | --- |
| | Unadjusted | | Adjusted* | | Adjusted* | |
| | β | (95% CI) | β | (95% CI) | β | (95% CI) |
| Fasting glucose[a] | **0.393** | (0.2–0.5) | **0.323** | (0.1–0.5) | | |
| 2-hour glucose[a] | **0.149** | (0.1–0.2) | | | **0.177** | (0.1–0.3) |
| Prepregnancy body mass index | | | | | | |
| Normalweight | 1 | | 1 | | 1 | |
| Overweight | **0.334** | (0.1–0.6) | 0.256 | (-0.1–0.5) | 0.275 | (-0.03–0.6) |
| Obesity | 0.409 | (-0.1–0.8) | -0.047 | (-0.5–0.4) | 0.067 | (-0.4–0.5) |
| Age | **0.027** | (0.0–0.1) | 0.019 | (-0.01–0.04) | 0.016 | (-0.01–0.1) |

* Models are adjusted for cohort, smoking, parity and treatment by the WHO-1999 criteria in addition.

[a]Not mutually adjusted. Significant values presented in bold.

in the native Norwegian population, and demonstrated an influence of maternal glucose on fetal growth trajectories [24]. A study by Dias et al. also supports our findings, showing that WHO-2013 criteria is associated with greater birthweight in Sri Lankan pregnant women [25]. Furthermore, studies from the Born-in-Bradford cohort showed that infants with South Asian origin have greater fat mass at birth despite their lower birthweight, explained by higher maternal glucose levels [26, 27]. Clinicians should be mindful that although ethnic minority women from these countries have a lower risk of delivering LGA or macrosomic neonates, elevated blood glucose levels affect fetal growth, particularly in terms of greater adiposity in the children.

The finding of a lower rate of caesarean section and operative delivery in women diagnosed by the WHO-1999 criteria may be partially explained by a treatment effect, as the results of

GDM testing were openly disclosed to caregivers and women at diagnosis. Clinical decisions such as timing of birth and induction of labour may be influenced by antenatally labelling pregnant women as having GDM. This could also be a possible explanation of lower rates of macrosomia observed in treated women in our study. Surprisingly, the expected beneficial effect of GDM treatment was not evident in the outcome of LGA, for which we found the highest risk among treated women. Mean gestational week at time of OGTT was 30 in our study, which may be too late for treatment to have a beneficial effect on fetal growth. Nevertheless, our results highlight the importance of identifying women with GDM to prevent complications and plan for a safer delivery in those with only moderately elevated fasting glucose as well, as these women demonstrate a higher risk of poor pregnancy outcomes, not observed in treated women with an established GDM diagnosis.

Our study has a number of strengths. We took advantage of previously collected maternal and offspring data, allowing more powerful and flexible analyses. Unlike many studies, participants were not selected based on high risk, as an OGTT was offered to all pregnant women. Our study also included women from some of the fastest-growing minority groups in Norway with a substantial risk of developing GDM. However, nearly all non-European women came from one study and the majority of Asians were of South Asian origin. As the overall race and ethnic composition in Europe differs and is constantly changing, the proportions included may not be fully representative for the present pregnant population.

Our study also has limitations. Glucose results were not blinded in the original studies and women with GDM were routinely treated when diagnosed by the WHO-1999 criteria. Information about adherence to any advice given and whether target glucose levels were achieved or not was unavailable. Therefore, any conclusions drawn about clinical outcomes should be interpreted with caution as treatment of GDM may be expected to lower the proportion of adverse outcomes and the natural effect of maternal hyperglycaemia. When comparing women with a GDM diagnosis by the WHO-2013 and Norwegian-2017 criteria, we have tried to control for this factor by adjusting for treatment. In addition, we performed sensitivity analysis excluding treated women which also showed similar associations of GDM and the studied outcomes. The rates of overweight and obesity in our cohort were somewhat lower than the general population, but closely approximated that of reproductive-aged women in Norway (8% vs 12% obesity nationally in 2018). Finally, we used a modified WHO-2013 criteria with only two timepoints, as 1-hour glucose concentrations were not collected in our study. The prevalence of GDM by the WHO-2013 criteria would presumably have been higher with the addition of the 1-hour timepoint.

## Conclusion

After accounting for important confounders, women retrospectively identified as having GDM according to the Norwegian-2017- or WHO-2013 criteria, but not diagnosed and treated by the WHO-1999 criteria, implying that they had moderately elevated fasting glucose only, had an increased risk of LGA, cesarean section and operative vaginal delivery when compared to women without GDM by any criteria. Identification of these women may enable caregivers to better plan for a safer delivery for both women and their infants. Our data support the use of a fasting glucose threshold corresponding to a twofold risk of adverse pregnancy outcomes, as using the Norwegian-2017 criteria would identify women at substantial risk for adverse outcomes without increasing the prevalence of GDM. What remains unanswered and can be established by randomized trials is whether treating mild fasting hyperglycaemia benefits women and their offspring, leading to an improvement in perinatal outcomes.

## Supporting information

**S1 File.**
(XLSX)

**S1 Fig. Flowchart of included studies and excluded participants from each study.** Study names listed in the top boxes. TRIP: Training in pregnancy.
(SVG)

## Acknowledgments

The authors would like to thank the following members of the Norwegian Hyperglycaemia in Pregnancy consortium: Marie Cecilie Paasche-Roland (Oslo University Hospital-Rikshospitalet, Oslo, Norway) for her contribution in the STORK Rikshospitalet study, Siv Mørkved (Norwegian University of Science and Technology, Trondheim, Norway) for her contribution in the TRIP study, and Ingvild Vistad (University of Bergen, Bergen, Norway) for her contribution in the Fit for Delivery study.

## Author Contributions

**Conceptualization:** Anne Karen Jenum, Nina Cecilie Øverby, Elisabeth Qvigstad, Linda Reme Sagedal.

**Data curation:** Signe Nilssen Stafne, Elisabeth Qvigstad.

**Formal analysis:** Anam Shakil Rai.

**Investigation:** Anne Karen Jenum, Linda Reme Sagedal.

**Methodology:** Line Sletner, Nina Cecilie Øverby, Are Hugo Pripp.

**Project administration:** Anam Shakil Rai, Line Sletner, Anne Karen Jenum, Linda Reme Sagedal.

**Supervision:** Line Sletner, Anne Karen Jenum, Nina Cecilie Øverby, Linda Reme Sagedal.

**Writing – original draft:** Anam Shakil Rai.

**Writing – review & editing:** Anam Shakil Rai, Line Sletner, Anne Karen Jenum, Nina Cecilie Øverby, Signe Nilssen Stafne, Elisabeth Qvigstad, Linda Reme Sagedal.

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
