## [Decision Letter · Decision Letter 0]

15 Mar 2022

PONE-D-22-04306Adverse pregnancy outcomes among women in Norway with gestational diabetes using three diagnostic criteriaPLOS ONE

Dear Dr. Rai,

Thank you for submitting your manuscript to PLOS ONE. After careful consideration, we feel that it has merit but does not fully meet PLOS ONE’s publication criteria as it currently stands. Therefore, we invite you to submit a revised version of the manuscript that addresses the points raised during the review process.

We look forward to receiving your revised manuscript.

Kind regards,

Andreas Beyerlein

Academic Editor

PLOS ONE

Journal Requirements:

4.We note that you have included the phrase “data not shown” in your manuscript. Unfortunately, this does not meet our data sharing requirements. PLOS does not permit references to inaccessible data. We require that authors provide all relevant data within the paper, Supporting Information files, or in an acceptable, public repository. Please add a citation to support this phrase or upload the data that corresponds with these findings to a stable repository (such as Figshare or Dryad) and provide and URLs, DOIs, or accession numbers that may be used to access these data. Or, if the data are not a core part of the research being presented in your study, we ask that you remove the phrase that refers to these data.

Additional Editor Comments:

In the spirit of Open and Reproducible Science, the analysis code should be made available in an online repository together with a data dictionary, and the respective URL should be mentioned in the Methods section.

Reviewers' comments:

Reviewer's Responses to Questions

**Comments to the Author**

1. Is the manuscript technically sound, and do the data support the conclusions?

Reviewer #1: Yes

Reviewer #2: Yes

2. Has the statistical analysis been performed appropriately and rigorously? 

Reviewer #1: Yes

Reviewer #2: Yes

3. Have the authors made all data underlying the findings in their manuscript fully available?

Reviewer #1: Yes

Reviewer #2: Yes

4. Is the manuscript presented in an intelligible fashion and written in standard English?

Reviewer #1: Yes

Reviewer #2: Yes

5. Review Comments to the Author

Reviewer #1: Dear authors,

Thank you very much and congratulation for a very well conducted study.

Please find bellow some comments for your consideration.

Overall, the article by Anam Shakil Rai and colleague is very clear and described a very well conducted epidemiological study. The author have tested how the classification according to criteria based on cut-off values for FG and 2HG may associated with the risk of perinatal outcome. The data are original and informative, and ,the results are clinically relevant for Public Health.

I have very little comments to the study in general as the sections are concise and well described showing very good skills from the team overall.

I would recommend the authors to make two clarifications:

The authors pooled 4 studies: 2 cohorts and 2 RCTs and have chosen to make a pooled analysis.

One sentence in the method section:

“Studies that only included specific subgroups (e.g. obese women only) or without the core data were not considered for inclusion” suggests that the number of studies to start with was chosen according to selection criteria. This could be clarified.

Can the authors justify why they have chosen to make a pooled analysis (adjusted for study cohorts) instead of making a meta-analysis. Although this second approach has less statistical power it may help observing study specific effects. Please justify and may be discuss in the strength and limitation section.

Although it is well explained in the main text, the sentence in the abstract: Asians had a lower risk of delivering large-for-gestational-age infants than Europeans […] can be misleading to readers with limited knowledge in birth weight categorisation. Although you informed (’but maternal glucose values were similarly positively associated with birthweight in all ethnic groups’) I think that there is still a risk of miss-interpretation.

In theory, as Asian babies are clearly off-chart when it comes to LGA and SGA categorization (based on the Norwegian scales) the analyses cannot be performed as they are meaningless. I would suggest to remove Asian from the LGA analyses.

The discussion is very clear and concise. However, all the comma separators have been substituted by dots. This should be corrected of course.

Reviewer #2: 1. The study is relatively well conducted.

2. The formatting leaves much to be desired - for some part is not consistent and a lot of full stop '.' which should be comma ','.

3. In Figure 1 what are 'STORK Groruddalen', 'STORK Rikshospitalet', 'Fitfor Delivery', & 'TRIP' - all these acronyms must be explained in the figure.

4. The standard nomenclature for International Association of Diabetes and Pregnancy Study Groups (IADPSG) and World Health Organization 2013 (WHO 2013) recommendations is IADPSG or WHO 2013.

Using 2013(superscript)WHO criteria is not standard way of describing. Please show why are these criteria depicted this way - 2013WHO and not WHO 2013 criteria.

5. The study must make it clear they are not using the full 3 point WHO 2013 or IADPSG but instead using a modified 2 point IADPSG or modified WHO 2013 criteria without 1 hour. The 1 hour time point is an essential part of the IADPSG criteria and will independently adds another one third of cases. This must be stated clearly a limitation that only fasting and 2 hour timepoints are used and thus they are not reflecting the IADPSG full criteria but only comparing the 2 time points criteria within the WHO 2013 criteria.

6. It was shown that Asians had a lower risk of delivering large-for-gestational-age babies. This is not helpful as Asians are smaller size and their babies size norms may be lower. Are customised charts used for Asians, e.g. in relation to mother height and weight and ethnicity?

7. However can these findings be applied to Asia in terms of the timepoints? e.g. fasting and relation to baby birthweights etc. Please refer to an Asian paper - Sri Lanka paper with criteria similar to Norwegian for fasting time point, where a similar aspect of changes in the criteria affect the number of cases

- Dias T, Siraj SHM, Aris IM, Li LJ, Tan KH. Comparing Different Diagnostic Guidelines for Gestational Diabetes Mellitus in Relation to Birthweight in Sri Lankan Women. Front Endocrinol (Lausanne). 2018 Nov 15;9:682. doi: 10.3389/fendo.2018.00682. PMID: 30524375; PMCID: PMC6262349.

6. PLOS authors have the option to publish the peer review history of their article (what does this mean?). If published, this will include your full peer review and any attached files.

Reviewer #1: No

Reviewer #2: No

---

## [Author Response · Author response to Decision Letter 0]

23 May 2022

Point-by-point response to reviewers

Reviewer 1: 

Comment: I would recommend the authors to make two clarifications:

The authors pooled 4 studies: 2 cohorts and 2 RCTs and have chosen to make a pooled analysis.

One sentence in the method section:

“Studies that only included specific subgroups (e.g. obese women only) or without the core data were not considered for inclusion” suggests that the number of studies to start with was chosen according to selection criteria. This could be clarified.

Reply: Thank you for your overall positive review and your valuable comments. We agree that we can provide clarity to this part. We have now added to this paragraph the following sentence: “The four studies were merged to perform a pooled analysis.” (page 5)

We agree that this sentence may be misinterpreted. We have now changed the sentence to “Only studies that had core data and were not based on specific selection criteria (e.g. obese women only) were considered for inclusion.”(p.5)

Comment: Can the authors justify why they have chosen to make a pooled analysis (adjusted for study cohorts) instead of making a meta-analysis. Although this second approach has less statistical power it may help observing study specific effects. Please justify and may be discuss in the strength and limitation section.

Reply: Thank you for suggesting an alternative statistical analysis. In principal we have performed statistical analysis comparable with an IPD-meta analysis. However, the analysis was not performed in relation with a systematic review, as the studies included were chosen based on a consortium of studies with glucose data. As statistical power was limited also in the pooled analysis, we did not consider presenting results for these relatively rare outcomes from each individual study (as often done as a first step in a meta-analyses). We did however use the same principles as in a meta-analysis for harmonization of variables etc, and a simpler search did not identify other Norwegian studies, expect one study of obese pregnant women. Nevertheless, as we had not strictly followed all methodology recommended for an IPDMA we chose to call this a "pooled analysis".

Comment: Although it is well explained in the main text, the sentence in the abstract: Asians had a lower risk of delivering large-for-gestational-age infants than Europeans […] can be misleading to readers with limited knowledge in birth weight categorisation. Although you informed (’but maternal glucose values were similarly positively associated with birthweight in all ethnic groups’) I think that there is still a risk of miss-interpretation.

Reply: This sentence could arguably be clearer, and we have now adjusted it as follows below. We hope that it will be acceptable that the abstract now consists of 301 words.

 “Asians had a lower risk of delivering large-for-gestational-age infants than Europeans, when applying national birthweight references, but maternal glucose values were similarly positively associated with birthweight in all ethnic groups". 

Comment: In theory, as Asian babies are clearly off-chart when it comes to LGA and SGA categorization (based on the Norwegian scales) the analyses cannot be performed as they are meaningless. I would suggest to remove Asian from the LGA analyses.

Reply: Thank you for your suggestion. We agree that not having an ethnicity-specific definition for LGA is a limitation, and this is also discussed in the paper. Unfortunately, there are no customized charts for this group in Norway. In our considered opinion, it is better to employ the national references, which are regularly used in clinical practice for all pregnant women living in Norway, than to completely remove the Asian group from the analysis. As LGA based on a Norwegian reference population is not a very precise measure for Asian women, we have additionally performed linear regression using birthweight z-score (table 4) to explore the association with maternal glucose.

Comment: The discussion is very clear and concise. However, all the comma separators have been substituted by dots. This should be corrected of course.

Reply: I am sorry for this error that appeared at submission. This have now been corrected. 

Reviewer 2: 

Comment: In Figure 1 what are 'STORK Groruddalen', 'STORK Rikshospitalet', 'Fitfor Delivery', & 'TRIP' - all these acronyms must be explained in the figure.

Reply: Thank you for your valuable comments and suggestions. These are actually not acronyms but the original study titles (Groruddalen and Rikshospitalet are places/hospitals in Norway). TRIP is an acronym for Training in pregnancy. We have added an explanation to the figure text to make this clearer. 

“Study names listed in the top boxes. TRIP: Training in pregnancy.” (p. 7)

Comment: The standard nomenclature for International Association of Diabetes and Pregnancy Study Groups (IADPSG) and World Health Organization 2013 (WHO 2013) recommendations is IADPSG or WHO 2013.

Using 2013(superscript)WHO criteria is not standard way of describing. Please show why are these criteria depicted this way - 2013WHO and not WHO 2013 criteria.

Reply: We chose superscript for the years (2013, 2017 and 1999) to increase readability and make it easier to follow for the reader, as these criteria are mentioned a large number of times and it can be confusing with all the numbers. 

Comment: The study must make it clear they are not using the full 3 point WHO 2013 or IADPSG but instead using a modified 2 point IADPSG or modified WHO 2013 criteria without 1 hour. The 1 hour time point is an essential part of the IADPSG criteria and will independently adds another one third of cases. This must be stated clearly a limitation that only fasting and 2 hour timepoints are used and thus they are not reflecting the IADPSG full criteria but only comparing the 2 time points criteria within the WHO 2013 criteria.

Reply: We agree that this is an important point that should be emphasized. It has been mentioned as a limitation in the discussion, but to provide even more clarity, we have added the points suggested by you to this paragraph.

 “Finally, we used a modified 2013WHO criteria with only two timepoints, as data for 1-hour glucose concentrations were not collected in our study. The prevalence of GDM by the 2013WHO criteria would presumably have been higher with the addition of the 1-hour timepoint.” (p. 22)

Comment: It was shown that Asians had a lower risk of delivering large-for-gestational-age babies. This is not helpful as Asians are smaller size and their babies size norms may be lower. Are customised charts used for Asians, e.g. in relation to mother height and weight and ethnicity?

Reply: See also response to reviewer 1. LGA was calculated using the same Norwegian reference population as used in clinical practice, for classification of LGA/SGA by sex and gestational age. Ethnically customized centile charts are not available in Norway (being a small country), and using customized charts from other countries could also potentially introduce bias, as mean birthweight in Norway is generally higher than for example UK (both for “White” and several immigrant groups). As we agree that LGA based on a Norwegian reference population is not a very precise measure for Asian women, we have instead performed linear regression using birthweight z-score (table 4) to explore the association with maternal glucose. We believe this analysis provides more useful knowledge for the non-European groups. 

Comment: However can these findings be applied to Asia in terms of the timepoints? e.g. fasting and relation to baby birthweights etc. Please refer to an Asian paper - Sri Lanka paper with criteria similar to Norwegian for fasting time point, where a similar aspect of changes in the criteria affect the number of cases

- Dias T, Siraj SHM, Aris IM, Li LJ, Tan KH. Comparing Different Diagnostic Guidelines for Gestational Diabetes Mellitus in Relation to Birthweight in Sri Lankan Women. Front Endocrinol (Lausanne). 2018 Nov 15;9:682. doi: 10.3389/fendo.2018.00682. PMID: 30524375; PMCID: PMC6262349.

Reply: thank for you suggesting this relevant paper. We have added this reference in the discussion, as it provides valuable insight into the subject discussed. 

“A study by Dias et al. also supports our findings, showing that 2013WHO criteria is associated with greater birthweight in Sri Lankan pregnant women. (p.21)

---

## [Decision Letter · Decision Letter 1]

28 Jun 2022

PONE-D-22-04306R1Adverse pregnancy outcomes among women in Norway with gestational diabetes using three diagnostic criteriaPLOS ONE

Dear Dr. Rai,

Thank you for submitting your manuscript to PLOS ONE. After careful consideration, we feel that it has merit but does not fully meet PLOS ONE’s publication criteria as it currently stands. Therefore, we invite you to submit a revised version of the manuscript that addresses the points raised during the review process.

We look forward to receiving your revised manuscript.

Kind regards,

Andreas Beyerlein

Academic Editor

PLOS ONE

Journal Requirements:

Additional Editor Comments (if provided):

- Suggest to use the abbreviation LGA also in the abstract.

- Abstract / Results: In case of a lower or upper limit of 1.0 in a 95% CI, please add as many digits as necessary to indicate whether the 95% CI contains the 1 or not. Mentioning p-values in the main text will then become unnecessary.

- Is it correct that those mothers identified by the 2013WHO and 2017Norwegian criteria but not diagnosed and treated by 1999WHO criteria were mothers with an FPG between 5.1 and 6.9 mmol/L, irrespectively of 2HG? If so, the authors might consider to mention this definition throughout the manuscript to describe this group instead of the lengthy wording used in the manuscript (after justifying it based on the different GDM definitions).

- "Distributions of all potential covariates were tested for normality." Which test was used? Were all continuous covariates assumed to follow a normal distribution based on the test results? Please add this information to the main text.

- Table 1: Please clarify that % values refer to column %.

- Results: ORs are sometimes given with 2, sometimes with 3 digits. This should be handled in a uniform manner, apart from 1.0 in 95% CIs as mentioned above.

- It is unclear to me why the sensitivity analysis was done and what it adds. Please explain this in some detail.

- Table 4:

* Were the conditions for applying a linear regression between maternal glucose and offspring's birthweight checked? In particular, is it plausible to assume a linear relationship instead of a J-shaped or U-shaped one?

* Presumably, FPG and 2HG were not mutually adjusted due to collinearity. This should be explained in the text.

* Asian, unadjusted: "P" should read "95% CI"

* The sign "-" should not be used as a separator between the lower and upper limit of a 95% CI, as it is already needed to indicate negative values. Suggest to use "," or ";" instead (consistently throughout the manuscript)

- In the spirit of Open and Reproducible Science, the analysis code (or SPSS syntax) should be made available in an online repository together with a data dictionary, and the respective URL should be mentioned in the Methods section. Can the data also be made available, and if not, why not?

Reviewers' comments:

Reviewer's Responses to Questions

**Comments to the Author**

1. If the authors have adequately addressed your comments raised in a previous round of review and you feel that this manuscript is now acceptable for publication, you may indicate that here to bypass the “Comments to the Author” section, enter your conflict of interest statement in the “Confidential to Editor” section, and submit your "Accept" recommendation.

Reviewer #2: All comments have been addressed

2. Is the manuscript technically sound, and do the data support the conclusions?

Reviewer #2: Yes

3. Has the statistical analysis been performed appropriately and rigorously? 

Reviewer #2: Yes

4. Have the authors made all data underlying the findings in their manuscript fully available?

Reviewer #2: Yes

5. Is the manuscript presented in an intelligible fashion and written in standard English?

Reviewer #2: Yes

6. Review Comments to the Author

Reviewer #2: Using a prefix, superscript year before the criteria is highly unconventional in medical literature for diabetes (usually reserved for atomic particles); and may also lead to formatting, spacing and search issues when published. Would still suggest in order of preference, for the 'year' , to be after the criteria and non superscript with space or dash or bracket if needed; or to be after the criteria as postfix.

7. PLOS authors have the option to publish the peer review history of their article (what does this mean?). If published, this will include your full peer review and any attached files.

Reviewer #2: **Yes: **Kok Hian TAN

---

## [Author Response · Author response to Decision Letter 1]

11 Aug 2022

Response to editor: 

Comment: Suggest to use the abbreviation LGA also in the abstract.

Reply: I have now added the abbreviation LGA to the abstract and use this instead.

Comment: Abstract / Results: In case of a lower or upper limit of 1.0 in a 95% CI, please add as many digits as necessary to indicate whether the 95% CI contains the 1 or not. Mentioning p-values in the main text will then become unnecessary.

Reply: As suggested, I have now removed the p-values from the main text and added extra digits to the 95 CI% with a lower or upper limit of 1.0. 

Comment: Is it correct that those mothers identified by the 2013WHO and 2017Norwegian criteria but not diagnosed and treated by 1999WHO criteria were mothers with an FPG between 5.1 and 6.9 mmol/L, irrespectively of 2HG? If so, the authors might consider to mention this definition throughout the manuscript to describe this group instead of the lengthy wording used in the manuscript (after justifying it based on the different GDM definitions).

Reply: This is not entirely correct. Mothers identified by the 2013-WHO and 2017-Norwegian criteria but not diagnosed and treated by 1999-WHO criteria were mothers with an FPG between 5.1 (or 5.3) and 6.9 mmol/L and with a 2H glucose < 7.8 mmol/L. To keep this as clear as possible to the readers we believe it’s necessary to refer to it in this way, although we agree it’s a bit lengthy. 

Comment: "Distributions of all potential covariates were tested for normality." Which test was used? Were all continuous covariates assumed to follow a normal distribution based on the test results? Please add this information to the main text.

Reply: The information about the tests is now added to the main text. “Assumptions for statistical analysis were tested and distributions of all potential covariates were checked for normality using Tests of Normality and by inspection of probability plots, which confirmed that these variables followed a normal distribution.” (p.7-8)

Comment: Table 1: Please clarify that % values refer to column %.

Reply: To clarify this we have now changed the legend for Table 1 to: 

Table 1. Characteristics and pregnancy outcomes in total study sample and according to their glucose tolerance status, using three diagnostic criteria for gestational diabetes. For each column data are presented as mean ± SD or n (%).

Comment: Results: ORs are sometimes given with 2, sometimes with 3 digits. This should be handled in a uniform manner, apart from 1.0 in 95% CIs as mentioned above.

Reply: Thank you for pointing this out. We have corrected this, and all OR’s are now given with 3 digits. 

Comment: It is unclear to me why the sensitivity analysis was done and what it adds. Please explain this in some detail.

Reply: In the majority of studies similar to ours, researchers exclude participants that have received an intervention/treatment. As much as we would like to do the same, we were unable to due to limited number of participants in our studies. Excluding all participants diagnosed and treated based on the 1999-WHO criteria would leave us with reduced power to explore the outlined associations. The sensitivity analysis was therefore mainly performed to verify the results achieved by the regression analyses where adjustments for treatment and a known diagnosis were made to the model as an alternative to excluding women. To further clarify this point, the following part has been added to the manuscript: 

“As a sensitivity analysis, and to verify the results achieved by the analyses where adjustment for treatment and a known diagnosis were made to the model, we repeated the same analysis after excluding participants who were diagnosed and treated based on the 1999WHO criteria as an alternative to excluding women.” (p.8-9). 

Comment: 

- Table 4:

Were the conditions for applying a linear regression between maternal glucose and offspring's birthweight checked? In particular, is it plausible to assume a linear relationship instead of a J-shaped or U-shaped one?

Reply: The conditions for a linear regression were checked, and the relationship between maternal glucose (fasting and 2HG) and birthweight z-score was linear. The sentence: “The conditions for a linear regression were checked, confirming a linear relationship between maternal glucose values and offspring birthweight” has been added to the methods section (p. 9).

* Presumably, FPG and 2HG were not mutually adjusted due to collinearity. This should be explained in the text.

Reply: It is correct that FPG and 2HG were not mutually adjusted and this is mentioned in the table text. However, it can be explained in the text as well. One sentence about this has now been added under Statistical Analyses: “We did not mutually adjust for FPG and 2HG due to collinearity.” (p.9)

* Asian, unadjusted: "P" should read "95% CI"

Thank you for this remark. It has now been corrected. 

* The sign "-" should not be used as a separator between the lower and upper limit of a 95% CI, as it is already needed to indicate negative values. Suggest to use "," or ";" instead (consistently throughout the manuscript)

We have now changed this sign into to the suggested “,” throughout the manuscript. 

- In the spirit of Open and Reproducible Science, the analysis code (or SPSS syntax) should be made available in an online repository together with a data dictionary, and the respective URL should be mentioned in the Methods section. Can the data also be made available, and if not, why not?

Reply: The reason for not being able to share data has been explained thoroughly upon submission under “Data Availability”. This is copy of our statement previously sent:

The datasets generated and/or analyzed during the current study are not publicly available due to the dataset containing potentially sensitive data. The editors can access data (in de-identified form) used in the manuscript, code book, and analytical code upon request. The project manager will contribute to the access being provided under appropriate conditions. However, research data for this publication include identifying health information subject to confidentiality. It is therefore not possible to share raw data publicly.

Reviewer #2: 

Comment: Using a prefix, superscript year before the criteria is highly unconventional in medical literature for diabetes (usually reserved for atomic particles); and may also lead to formatting, spacing and search issues when published. Would still suggest in order of preference, for the 'year' , to be after the criteria and non superscript with space or dash or bracket if needed; or to be after the criteria as postfix. 

Reply: Thank you for your suggestion. We have changed this throughout the manuscript, and the criteria are now listed as: WHO-1999, WHO-2013 and Norwegian-2017.

---

## [Decision Letter · Decision Letter 2]

4 Oct 2022

PONE-D-22-04306R2Adverse pregnancy outcomes among women in Norway with gestational diabetes using three diagnostic criteriaPLOS ONE

Dear Dr. Rai,

Thank you for submitting your manuscript to PLOS ONE. After careful consideration, we feel that it has merit but does not fully meet PLOS ONE’s publication criteria as it currently stands. Therefore, we invite you to submit a revised version of the manuscript that addresses the points raised during the review process.

 More specifically, please address all comments from the reviewer 3, and detail what/where changes are made. . 

We look forward to receiving your revised manuscript.

Kind regards,

Zhong-Cheng Luo

Academic Editor

PLOS ONE

Reviewers' comments:

Reviewer's Responses to Questions

**Comments to the Author**

1. If the authors have adequately addressed your comments raised in a previous round of review and you feel that this manuscript is now acceptable for publication, you may indicate that here to bypass the “Comments to the Author” section, enter your conflict of interest statement in the “Confidential to Editor” section, and submit your "Accept" recommendation.

Reviewer #2: All comments have been addressed

Reviewer #3: (No Response)

Reviewer #4: All comments have been addressed

Reviewer #5: All comments have been addressed

2. Is the manuscript technically sound, and do the data support the conclusions?

Reviewer #2: Yes

Reviewer #3: Partly

Reviewer #4: Yes

Reviewer #5: Yes

3. Has the statistical analysis been performed appropriately and rigorously? 

Reviewer #2: Yes

Reviewer #3: Yes

Reviewer #4: I Don't Know

Reviewer #5: Yes

4. Have the authors made all data underlying the findings in their manuscript fully available?

Reviewer #2: Yes

Reviewer #3: No

Reviewer #4: Yes

Reviewer #5: Yes

5. Is the manuscript presented in an intelligible fashion and written in standard English?

Reviewer #2: Yes

Reviewer #3: Yes

Reviewer #4: Yes

Reviewer #5: Yes

6. Review Comments to the Author

Reviewer #2: The authors have make amendments to my suggestion of not using prefix for GDM criteria. No more comments.

Reviewer #3: See attached file

The authors present the results of a cohort study that aimed to compare perinatal outcomes between women diagnosed and treated for GDM by the more stringent WHO 1999 diagnostic criteria and women who during that period would have theoretically been diagnosed as having GDM by the lower threshold 2013 and 2017 criteria but were hence not labelled as GDM and thus untreated. This subject has been previously assessed in other studies in different countries and with varying combinations of GDM diagnostic thresholds. The cohort was somewhat unusually constructed by pooling data from 2 cohort studies and two RCTs which might insert an unmeasurable selection bias and influence generalizability. I am not sure whether this a revision of a previously submitted manuscript, but I have not had the opportunity to review it previously.

Overall, this a well thought out paper that is providing some additional information to the many papers that have been published attempting to retrospectively analyze the perinatal impact of changing GDM diagnostic criteria.

Below are my itemized individual comments and questions:

Reviewer #4: The main query is understanding the groups in the WHO 2013 and Norwegian 2017 cohorts that were offered treatment. In Table 1 e.g the WHO 2013 column for the non-GDM has 119 and for GDM 199. From the text it is stated that they were offered treatment if they met the WHO 1999 criteria . Presumably this meant that their 2-hr levels were between 7.8 and 8.5 mmol/l for those that were "non-GDM" and ≥8.5 for the "GDM" group? If this is a correct interpretation it would help to have this spelt out more clearly. The dilemma with labelling them as WHO 2013 GDMs is as you have noted that it misses the group who would have been diagnosed by the 1-hr value of ≥10.0.

My recommendation is that including this group does not add significantly to the paper and could be removed.

Table 1Normal weight should be ≤ 24.9 not ≥24.9

Minor correction the HAPO study being published in the USA is hyperglycemia, not hyperglycaemia.

Reviewer #5: (No Response)

7. PLOS authors have the option to publish the peer review history of their article (what does this mean?). If published, this will include your full peer review and any attached files.

Reviewer #2: **Yes: **Kok Hian TAN

Reviewer #3: No

Reviewer #4: **Yes: **Jeremy J N Oats

Reviewer #5: No

---

## [Author Response · Author response to Decision Letter 2]

13 Dec 2022

The point-by-point response to reviewers is attached to this submission. 

Reviewer 3

1. Abstract, results: The term “women diagnosed with GDM by all three criteria” might be confusing. Do you mean that women that would have been considered positive by either of the three diagnostic criteria i.e all women with either FPG>5.0 or 2hr >7.7 which would infer pooling of the GDM women in the cohorts or is the intent to say that women who were diagnosed in either of the groups (non-pooling) had increased rates of LGA. I assume the latter as this is more in line with what is stated in the results. 

Respond: Thank you for your thorough review and overall positive comments. 

We agree that the wording can be confusing, and as you suggest, we have edited the statement to: 

“Compared to the non-GDM group, women diagnosed with GDM by either of the three criteria had an increased risk of large-for-gestational-age infants (adjusted odds ratios (OR) 1.7-2.2).” (p.2)

2. Introduction 1st paragraph: I would reconsider stating that “Gestational diabetes mellitus (GDM) is associated with … foetal and neonatal mortality”. While there is good data to support the association with other perinatal outcomes, this is not true for stillbirth and neonatal death. The reference provided (1) does not support this statement. 

Respond: Foetal and neonatal mortality is now removed from the sentence, and the reference is updated to ensure that the content corresponds with the phrase. The new sentence is: 

“Gestational diabetes mellitus (GDM) is associated with increased risk of macrosomia, caesarean section, preeclampsia and preterm delivery. (p.4) 

3. Methods: Justify the grouping of south and east Asian ethnicity under “Asian”. Do these two distinct ethnic backgrounds have similar BMI/insulin resistance levels and rate of GDM related outcomes? Many studies have suggested that South Asians are a distinct group with regards to glucose handling. 

Respond: This is a good point. However, out of the 223 Asian women included in this study, only 39 had East Asian origin. We did not have statistical power to study the relationship between glucose levels and the related outcomes in these two groups separately, but sensitivity analyses (not shown) did not suggest that they differed. We therefore made the decision to combine them. The mean BMI and mean birthweight were quite similar in the two groups, but, not surprisingly, South Asians had a somewhat higher prevalence of GDM (42% vs 24% in East Asians, using 2013-WHO criteria). This is already added as a limitation in the discussion. 

“However, nearly all non-European women came from one study and the majority of Asians were of South Asian origin.” (p.25) 

4. Methods: I would like to understand why fetal deaths that occurred after glucose testing were excluded and not reported as an adverse outcome? This potentially removes some of the most severe cases of poor glycemic control. 

Respond: Fetal deaths in this study include abortion<22 weeks, stillbirth≥22 weeks and neonatal death. The majority of these occurred before the first OGTT, and we lack outcome data (birthweight, birth complications, sex etc.) from most of these. After excluding those lacking data on birthweight, we were left with only one fetal death (with normal glucose values) which was then excluded. We have changed the order of the words in the sentence to make it clearer that infants with missing birthweight were excluded first, and then fetal deaths (and figure 1 shows that only one fetal death was excluded). 

“After excluding women with multiple pregnancies, those lacking glucose values, infants with missing birthweight and foetal deaths, the study sample consisted of 2970 mother-child pairs.” 

5. I am trying to understand how many women diagnosed with GDM by WHO -99 criteria failed lifestyle modification. This is not reported in Table 1 but in the methods it is mentioned that only 12 needed “such pharmacological treatment” which I will assume means either insulin or oral hypoglycemics (correct me if I am wrong). If this is the case, then the rate of lifestyle modification failure was a bit less than 3.8%. This is much less than the reported rate in the literature (20- 40%) and is even more surprising in light of the fact that the FPG threshold for diagnosis and treatment was > 6.9 mmol/L which would have led to a high probability of treatment failure. When examining the mean FPG in the WHO-99 GDM group, the mean FPG was 5.0+/- 0.6 thus it seems that there were very few women with significantly elevated FPG raising questions regarding how representative this cohort is of the GDM population in general ( see comment my comment in the preamble). Perhaps add categorical data on the FPG and 2 hr values (i.e FPG 5.1-5.5; 5.6-6.0; 6.0-6.4; 65-6.9) for clarification of the population characteristics.

Respond: Your understanding is correct regarding how many women failed lifestyle modification - only 12 women needed insulin or metformin, based on available information in the hospital records. Several points may influence this rather low number. Some women may have started with lifestyle modification but later needed medication, in which case their medication may have been overlooked and their treatment registered incorrectly. Furthermore, universal screening was performed in our four cohorts as opposed to risk-factor based screening often carried out in many comparable studies. With that, we include women with a milder form of GDM, and our study may be more representative of real-life low risk populations. We have also touched upon in the discussion that our study included women with less overweight/obesity than the general population which may also have a certain impact. 

6. Results: It is stated that “Women diagnosed with GDM by any criteria had a higher rate …”. I do not see a group defined as meeting ANY of the criteria thus this result is not visible in any of the tables and not indicated in the planned analysis (similar to comment 1 above). Please ensure that these group comparisons confirm to the planned analysis and to the groupings presented in the tables. 

Respond: The term “diagnosed with GDM by any criteria..” means any of the three criteria. To clarify this further, we have changed the sentence to: 

“All three groups of women diagnosed with GDM (according to WHO-1999, WHO-2013 and Norwegian-2017 criteria respectively) had a higher rate of LGA neonates compared to their non-GDM counterparts, while higher rates of macrosomia (birthweight>4000g) were only found in those diagnosed by Norwegian-2017 and WHO-2013 criteria.”(p.9)

7. Figures: pay attention to formatting issues. Figure 1 – not sure is needed in the main paper – could be in supplementary data. I would prefer that figure 1 be a diagram of the different cohorts by diagnostic criteria and the number that would have been identified and treated in each. 

Respond: We accept your suggestion of moving Figure 1 to supplementary data. As we mainly look at these four cohorts as one pooled dataset, our main focus is not showing GDM rates in each cohort etc. However, the data on number of women identified and treated in each cohort was published in the first paper from this consortium, and a reference to this specific paper for those interested in more information is already added, and can be found under number 12:

“Detailed study methods for the pooled data set have been previously described (12)..” (p.5)

8. Table 2: Not clear what the first column represents. Add heading. I would also make it clear in the title (and text) that there is a global risk reported for each outcome followed by a stratified analysis by BMI and ethnicity. In the footer define what BMI define Normal, overweight and obesity. Also – in the footer it says that the variables adjusted for are age. gestational weeks at delivery, parity and study cohort while in the text it states that there was also adjustment for treatment by the WHO-1999 criteria. Please clarify and ensure uniformity in reporting. 

Respond: Thank you for pointing out that this table was difficult to interpret; we appreciate the opportunity to clarify. The first column shows crude (unadjusted) analyses, while the next columns show adjusted analysis for each criteria (not a stratified analysis by BMI and ethnicity). This is now clearly presented in the table. The definition of BMI categories is now added to the footer.

Further, the analyses are also adjusted for treatment by WHO1999, as stated in the text and shown in the table, as well as pre-pregnancy BMI and ethnicity. The footnote stated that the analysis was “Additionally adjusted for age, maternal smoking, parity and study cohort” – these are the variables we adjusted for but which are not shown separately in the table. The actual number for the rest (treatment, BMI etc) are listed in the tables. 

In order to avoid any confusion, we have amended the footer to explicitly define the adjusted analysis. The first applies to the column regarding WHO 1999 criteria, while the second applies to the WHO-2013 and Norwegian-2017 criteria.

* Adjusted for BMI group and ethnicity, as shown. Additionally adjusted for age, gestational weeks at delivery, parity and study cohort. 

∞Adjusted for GDM diagnosed by WHO-1990 criteria, BMI group and ethnicity, as shown. Additionally adjusted for age, gestational weeks at delivery, parity and study cohort.

New title for table 2: Crude and adjusted analyses of risk of large-for-gestational-age, total cesarean section and operative vaginal delivery, by GDM-criteria. (p.13). 

9. In both Table 2 and 3: I am not sure why under the outcome (LGA, CS etc) there is a subcategory of the diagnostic criteria. This is confusing and unnecessary as this is self-evident from the column identification. Also, Table 3 formatting is unusual with different Tables for each outcome and separate footnotes. Should align with format of Table 2. 

Respond: Table 2: the subcategory of diagnostic criteria is listed mainly to show the crude analyses for each criteria, and the unadjusted risk would be impractical to present otherwise. In table 3 we agree that we could have just presented one row and typed “GDM” instead of GDM and the respective criteria. We have also merged the three tables into one and placed the footnotes in the end of the table to align with the format of table 2. 

10. Also, in the title of Table 3 (and in text) it would be good to spell out for the reader that this analysis is for the subgroup of those that would have GDM by the newer criteria but were UNDIAGNOSED AS GDM AND THUS UNTREATED. 

Respond: Thank you for this suggestion. In this sensitivity analysis, we repeat the same analysis as in table 2, but after excluding women who were diagnosed and treated according to the WHO-1999 criteria. One could of course turn it around and give it a title as suggested. However, what we wish to highlight here is that this is the exact same analysis as table 2 (in which we adjusted for the treatment effect) but it is performed after excluding diagnosed/treated women. The analysis presented in Table 3 was performed mainly for statistical reassurance, to support and verify our main analysis (Table 2). 

11. As I was re-reading, another possibility is to add to Table 2 the untreated aOR with a footnote explaining the different cohort structure. This would allow the reader to compare visually the untreated vs the combination of treated and untreated cohorts. With appropriate Table design this could be not too complex. 

Respond: Thank you for your suggestion. This could be possible, but, as this table is already complex with a lot of data and details, we would prefer to keep it as it is. 

12. If needed – Table 4 could easily be a supplementary table. 

Respond: We would like to include this table to demonstrate the important point that the use of LGA as an outcome representing excessive fetal growth is challenging, as the definition of LGA is not ethnicity-specific. Thus, very few ethnic Asian babies were born LGA. With the help of the analyses presented in this table, we show that the effect of elevated glucose on birthweight was similar across ethnic groups. 

13. Discussion: Is the statement “Those retrospectively identified by the Norwegian-2017 and WHO-2013 criteria, but who were not diagnosed and treated by the WHO-1999 criteria (i.e., women with moderately elevated fasting glucose only) also had an increased risk of cesarean section and operative vaginal delivery after adjustment for confounders.” Accurate? In Table 3, that shows the results for “untreated GDM”, ONLY LGA was significant. Clarify and adjust discussion as needed. 

Respond: Our main analyses are presented in table 2 and our goal was to include all women without having to exclude a large number of participants as this might lead to power limitations and less representativity. However, with advice from study statistician and suggestions from reviewers in the first revision round, we included a sensitivity analysis (table 3) to support and verify the results of our main analysis. This is explained under the heading “statistical analysis”: 

“As a sensitivity analysis, and to verify the results achieved by the analyses where adjustment for treatment and a known diagnosis were made to the model, we repeated the same analysis after excluding participants who were diagnosed and treated based on the WHO-1999 criteria.” (p.8)

This sensitivity analyses gave almost identical effect estimates as the main analyses (table 2), although, when women with GDM by WHO-1999 were excluded, statistical significance was only reached for LGA. This was probably due to the decreased population size in this analysis, which is why our conclusion is based on the analyses in table 2 to a greater extent. 

14. At the end of the first paragraph in the discussion you state that “moderately elevated fasting glucose, when unidentified and untreated, is associated with several adverse outcomes” and that the increased risk was “…primarily based on elevated 2HG values.”. I do not see any separate analysis showing the effect of the individual abnormal 75 gram OGTT value except for the effect on z score birthweight which is definitely not an “adverse outcome” in itself. New results should not be entered into the discussion. Please clarify and if needed adjust manuscript accordingly. 

Respond: 

We regret that our wording has been unclear. Our intention was not to enter new results in this part of the paper, but rather to underline that the WHO-1999 criteria are primarily based on elevated 2HG values. To avoid misunderstandings, we have added to the sentence the words “which are”. 

“Taken together, these findings indicate that moderately elevated fasting glucose, when unidentified and untreated, is associated with several adverse outcomes that were not observed in women where GDM was detected and treated according to WHO-1999 diagnostic criteria, which are primarily based on elevated 2HG values. (p.19)

15. When comparing the results to what is known in the literature you cite studies from Canada. For your interest one of the first studies from Canada is Am J Obstet Gynecol 2015;212:224.e1-9.

Respond: Thank you for making us aware of this relevant paper. We have now added this study along with the other studies from Canada in the reference list. 

16. The last sentence in the conclusion states that “What remains unanswered and can be established by randomized trials is whether treating mild hyperglycaemia benefits women and their offspring, leading to an improvement in perinatal outcomes” is a bit bombastic and does not take into consideration ACHOIS (to a lesser degree) and the NICHD study (N Engl J Med 2009; 361:1339-1348).I would consider changing this. 

Respond: You are correct that if read alone, this sentence doesn’t take into consideration the studies of Crowther and Landon. However, this sentence is connected to the previous sentences and must be read in light of them. Our data demonstrates that women with moderately elevated fasting glucose levels have an increased risk of adverse pregnancy outcomes, but we need more data, and especially RCT’s, demonstrating whether treating women falling into this particular group (moderately elevated fasting glucose but a low 2HG value) has clinical value. Although both studies (ACHOIS and NICHD) demonstrated a benefit of treatment of GDM, both of these studies employed criteria focused on elevated post-load glucose values (ACHOIS used criteria similar to WHO 1999, and NICHD included only women with fasting glucose <5,3 mmol/l combined with elevated 1-, 2- or 3- hour values) thereby demonstrating the need for more information on the effect of treating women with moderately elevated fasting glucose values only. 

We have now modified this sentence to make it more nuanced: 

“What remains unanswered and can be established by randomized trials is whether treating mild fasting hyperglycaemia benefits women and their offspring, leading to an improvement in perinatal outcomes.”

Reviewer 4:

Reviewer #4: The main query is understanding the groups in the WHO 2013 and Norwegian 2017 cohorts that were offered treatment. In Table 1 e.g the WHO 2013 column for the non-GDM has 119 and for GDM 199. From the text it is stated that they were offered treatment if they met the WHO 1999 criteria . Presumably this meant that their 2-hr levels were between 7.8 and 8.5 mmol/l for those that were "non-GDM" and ≥8.5 for the "GDM" group? If this is a correct interpretation it would help to have this spelt out more clearly. The dilemma with labelling them as WHO 2013 GDMs is as you have noted that it misses the group who would have been diagnosed by the 1-hr value of ≥10.0.

My recommendation is that including this group does not add significantly to the paper and could be removed.

Table 1Normal weight should be ≤ 24.9 not ≥24.9

Minor correction the HAPO study being published in the USA is hyperglycemia, not hyperglycaemia.

Respond: Thank you for your comments and suggestions. 

It is correct that women in the study were offered treatment if they met the WHO 1999 criteria – that is, fasting glucose ≥7.0 mmol/l and/or 2-hour glucose (2HG) ≥7.8 mmol/l. In the analyses we have adjusted for the treatment effect (or excluded treated women in the sensitivity analyses, table 3). 

Women with GDM by WHO-2013 criteria, after controlling for or excluding treatment by WHO-1999 criteria, are those with fasting glucose 5.1-6.9 mmol/l and 2-hour glucose 2HG <7.8 mmol/l. 

Women with GDM by Norwegian-2017 criteria, after controlling for or excluding treatment by WHO 1999-criteria, are those with fasting glucose 5.3-6.9 mmol/l and 2-hour glucose <7.8 mmol/l. 

We have tried to explain this in the paper, both under statistical analyses and under results, see page 8 and 13:

“Doing so allowed us to identify the group of women with an elevated fasting blood glucose only (fasting glucose 5.1-6.9 mmol/l and 2HG <7.8 mmol/l for WHO-2013, and fasting glucose 5.3-6.9 mmol/l and 2HG <7.8 mmol/l for Norwegian-2017 criteria) who were untreated.”

“After adjusting for confounders and for treatment by the WHO-1999 criteria (thereby expressing the risk related to having fasting glucose 5.1-6.9 mmol/l while 2HG <7.8 mmol/l) the OR for LGA for this group was 1.70 (95% CI 1.2,2.5,) compared to non-GDM women (both fasting glucose <5.1 mmol/l and 2HG <7.8 mmol/l).(p.13)” etc

The reason we didn’t exclude treated women completely from our analyses is that it would affect representativity and result in low power to detect associations with GDM. Therefore, we have instead adjusted for the treatment effect when studying this group. 

It is unclear if you suggest removing the women diagnosed by WHO 2013 criteria, as we do not have information on women with elevated 1-hour glucose values. We maintain that it is important to include this group, despite our acknowledged limitation, as it allows us to compare the outcomes associated with two different thresholds for fasting glucose: 5.1 mmol/l for WHO 2013 and 5.3 mmol/l for Norwegian 2017. Of these, the WHO 2013 criteria are most widely used, and therefore relevant for comparison.

The symbols have now been corrected as per your suggestion, so is the spelling of “hyperglycemia”.

---

## [Editor Report · Decision Letter 3]

8 Jan 2023

Adverse pregnancy outcomes among women in Norway with gestational diabetes using three diagnostic criteria

PONE-D-22-04306R3

Dear Dr. Rai,

We’re pleased to inform you that your manuscript has been judged scientifically suitable for publication and will be formally accepted for publication once it meets all outstanding technical requirements.

Kind regards,

Zhong-Cheng Luo

Academic Editor

PLOS ONE
---

## [Editor Report · Acceptance letter]

12 Jan 2023

PONE-D-22-04306R3 

Adverse pregnancy outcomes among women in Norway with gestational diabetes using three diagnostic criteria 

Dear Dr. Rai:

I'm pleased to inform you that your manuscript has been deemed suitable for publication in PLOS ONE. Congratulations! Your manuscript is now with our production department. 

Kind regards, 

on behalf of

Dr. Zhong-Cheng Luo 

Academic Editor

PLOS ONE